# Management of Marine Natura 2000 Sites as Exemplified by Seabirds Wintering in the Baltic Sea: The Case of Poland

**Dominik Marchowski** [1,*] **, Łukasz Ławicki** [2] **and Jacek Kaliciuk** [3]

1   Ornithological Station, Museum and Institute of Zoology, Polish Academy of Sciences, Nadwiślańska 108, 80-680 Gdańsk, Poland
2   Eco-Expert, 70-206 Szczecin, Poland
3   West Pomeranian Nature Society, 71-415 Szczecin, Poland
*   Correspondence: dmarchowski@miiz.waw.pl

**Abstract:** Based on the example of wintering waterbirds in the Baltic Sea, we show an approach that is useful in defining priority species for management. The *Value Factor* (VF) is the quantitative method for evaluating the importance of an area for a species. Every year, 4,400,000 waterbirds winter in the Baltic. Among these, the highest priority species are velvet scoter *Melanitta fusca* (hereafter VS, VF = 153) and long-tailed duck *Clangula hyemalis* (hereafter LTD, VF = 204): 74% and 40%, respectively, of the world's populations, and over 90% of the EU populations of both species spend the winter in the Baltic. Management plans (hereafter MP) regulating the protection of marine Natura 2000 sites (hereafter MPA) and dedicated to the protection of VS and LTD have been implemented in 65% and 51%, respectively, of MPAs in the Baltic. Poland, a key country for the survival of these species, has not implemented a single MP despite the existence of documentation confirming their crucial importance for seaducks, and the pressures occurring there. We suggest using the VF concept to define priority species. On this basis, it will be possible to identify gaps in the protection of the most seriously threatened species and implement conservation measures at the most appropriate sites.

**Keywords:** seaducks; Natura 2000; velvet scoter; long-tailed duck; management of threatened species; biodiversity strategy

## 1. Introduction

Biodiversity loss has become one of the major causes of change in the Earth's ecosystems [1,2], which is why it is also becoming a threat to humanity [3]. In response to the extinction of species, an international coalition of scientists, conservationists, non-profits, and public officials–Nature Needs Half–was established, aiming to protect 50% of the Earth by 2030 (https://natureneedshalf.org/accessed on 5 September 2022). This coalition is an informal body aimed at raising awareness and influencing decisions made by governments. In Europe, there is a formal network—Natura 2000 [4]—created under the auspices of European Union (EU) law [5], which is now among the largest international networks of protected areas [6,7]. It can be successfully expanded towards more effective operations and into areas beyond Europe.

The EU pays special attention to the protection of biodiversity: directives intended for the consistent protection of entire ecosystems are developed based on long-term work by expert groups [8–10]. This comprehensive notion of protecting and managing the environment based on scientific criteria has led to the establishment of more than 26,000 protected sites covering about 26% of the land and 11% of the seas in the EU [11]. The two principal directives regulating the creation and management of these two independent networks are the Birds Directive (Council Directive 2009/147/EC on the conservation of wild birds—hereafter BD) and the Habitats Directive (Council Directive 92/43/EEC on the Conservation of natural habitats and of wild fauna and flora—hereafter HD). Under BD, Member States (hereafter MS) are obliged to create a separate network of areas for bird

protection called Special Protection Areas (hereafter SPA). HD, in turn, serves to protect all species of animals other than birds and plant habitats as Sites of Community Importance (hereafter SCI) and, after approval by the European Commission (hereafter EC), as Special Areas of Conservation [10].

Recently, in response to the climate crisis and biodiversity loss, the EC decided to take more ambitious steps to protect nature in Europe. In line with The European Green Deal [12], the EU's Biodiversity Strategy for 2030 [11] aims to expand the network to cover at least 30% of Europe's land and sea areas. In 2022, the EC presented guidelines with the expectation that the MS is supposed to implement them voluntarily by 2023 [13]; otherwise, the EC will consider European enforcement legislation by 2024 [11].

Birds are one of the best indicators of environmental quality [14]: trends among birds are used to interpret changes in the environment [15]. Global analyses undertaken on waterbirds clearly demonstrate the dependence of nature conservation on the effective governance of a country: the more effective the governance, the larger the area protected and hence the greater the number of waterbirds [16].

Here, we would like to discuss some aspects of the Natura 2000 network that could improve its functioning, just as Amano et al. [16] used waterbirds as an example to highlight certain shortcomings of this protection system. Moreover, bearing in mind the EU's aim to expand Marine Protected Areas (hereafter MPA) to make their protection comparable to land areas [11], we focused on marine SPAs, and more specifically on the marine birds for which such areas should be established. Europe is an important wintering ground for seabirds [17], which is why we shall analyse the effectiveness of the Natura 2000 network in relation to this group, focusing on the areas where they concentrate in the largest numbers.

We will focus on the Baltic Sea because it is one of the world's most important wintering sites for waterbirds breeding in the Arctic and high latitudes [18]. Therefore, what is happening in the Baltic region has a major impact on the entire global or flyway populations of several species of waterbirds. Extensive shallow banks, lagoons, and bays rich in benthic organisms provide ideal conditions for these birds [19]. While the Baltic Sea is a relatively important breeding ground for waterbirds, it is their concentrations during the non-breeding period that make it a unique place [17]. Hundreds of thousands of waterbirds gather here in this relatively small area to spend the winter [18].

While the Baltic Sea is important as a wintering ground for a wide group of waterbirds, the largest of these is comprised of seaducks—67% of all the waterbirds (excluding gulls) present in this body of water [18]. Seaducks spend their non-breeding period almost exclusively in marine waters, usually forming dense concentrations [20]. This adaptation has turned out to be an evolutionary success: these ducks are widespread and numerous throughout the northern hemisphere, with numbers estimated at ca 17,000,000 individuals [21]. Currently, however, human pressure on shelf seas is exposing this group of birds disproportionately to mass mortality [22]. Threats include the movement of ships, water sports and offshore wind farms [23]. However, the biggest danger of all comes from fisheries, with bycatch being one of the two most important threats to seabirds worldwide [24]. In the case of seaducks, bycatch in gillnet fisheries is the most important anthropogenic factor causing mortality [25,26]. As a result of the decrease in their numbers, 46% of seaduck species have been classified as threatened or near threatened [27]. In regions such as the Baltic Sea, where concentrations of wintering seaducks coincide with gillnet fisheries, there is a conflict between this and nature conservation objectives [26]. In such hot spots, it is important to apply an appropriate management approach with the aim of reconciling conflicting interests [28].

The basis for the conservation of species and their habitats in European Natura 2000 sites is the Management Plan (hereafter MP), a document that also establishes the legal basis for implementing the measures it sets out (Council Directive 2009/147/EC, Nature Protection Act, Journal of Laws 2018, item 1614). Without an MP, species and habitat protection are greatly limited if not absent altogether [29]. We conducted an analysis of all the marine Natura 2000 sites (SPAs only) established to protect waterbirds during the

non-breeding period (n = 117) in the entire Baltic Sea and checked whether each site has an MP. Those Exclusive Economic Zones (hereafter EEZ) in the Baltic sub-region with the lowest level of MP implementation were identified and the relevant documents were analysed in greater detail.

The basic criteria for classifying a site as species-critical (for SPA) are not the same in all EU countries. Some states apply biogeographic populations as references, others use national population numbers. Such an inconsistent approach causes problems in identifying priorities in the conservation of species most in need. In this study, we try to systematise this issue by introducing a simple indicator—*Value Factor*—which informs about the importance of a given area for the survival of a given species. Here, we wish to highlight the importance of the Baltic Sea's ecosystems for wintering species, and whether these are effectively protected.

## 2. Materials and Methods

### 2.1. Number of Wintering Waterbirds on the Baltic Sea

This study was carried out in the Baltic Sea and covered an area of 377,000 km$^2$ (Figure 1). We determined the percentage of the wintering waterbird population (excluding gulls) in the Baltic Sea in relation to the global and flyway populations based on available sources, articles, and books [18,22,30–32], documentation of management plans [33–35] and publicly accessible database [21]. The waterbird species analysed here are from the orders Anseriformes, Gaviiformes and Podicipediformes, the family Alcidae in the order Charadriiformes and the cormorant *Phalacrocorax carbo*. A seaduck group was separated from the other waterbirds and defined as follows: ducks that, during the non-breeding period, occur mainly at sea or on large inland lakes, but are absent from or appear only exceptionally on small inland bodies of water. They often congregate in large flocks. The group defined in this way includes the Anatidae subfamily, tribe Mergini (long-tailed duck *Clangula hyemalis*, Steller's eider *Polysticta stelleri*, spectacled eider *Somateria fischeri*, king eider *Somateria spectabilis*, common eider *Somateria mollissima*, surf scoter *Melanitta perspicillata*, velvet scoter *M. fusca*, siberian scoter *M. stejnegeri*, white-winged scoter *M. deglandi*, common scoter *M. nigra*, black scoter *M. americana*, red-breasted merganser *Mergus serrator* and harlequin duck *Histrionicus histrionicus*) and tribe Aythyini (greater scaup *Aythya marila*).

### 2.2. Assessing the Value of the Baltic Sea

Site-based conservation of birds has been used for a long time [36]; the determination of protected areas is based on a widely agreed upon set of international criteria consisting of the assessment of the size of the population inhabiting a given area [37].

To determine the value of the Baltic Sea waters for a given species, we multiplied the percentage of its population present in this region [18] by its IUCN conservation status code [38]: LC = 1, NT = 2, VU = 3, EN = 4, CE = 5. In this way, we obtained a *Value Factor* (VF) for the flyway and global populations. The larger the VF, the greater the importance of a given area for a particular species. Since both global and flyway population estimates usually lie within a certain range, we obtained six VFs for each species by multiplying the IUCN code by the mean, minimum, and maximum percentage in relation to the flyway and global populations. Thus, considering both variability of numbers and the value of the area on the flyway and at the global scale, VF is the average of these six values.

### 2.3. Effectiveness of Natura 2000 Sites for Protecting the Species with the Highest VF

The analysis covered all marine Natura 2000 sites, established on the basis of the Birds Directive to protect waterbirds during the non-breeding period in the Baltic Sea area (Figure 1, n = 117). The Standard Data Forms (SDF) of these sites were analysed (documentation accessed at https://natura2000.eea.europa.eu/accessed on 5 September 2022). They are situated in the following countries: Denmark, Germany, Poland, Lithuania, Latvia, Estonia, Sweden, and Finland. We checked whether marine Natura 2000 Special

Protected Areas (SPA) had been established for the migrating and wintering species and whether MPs were being implemented at these sites.

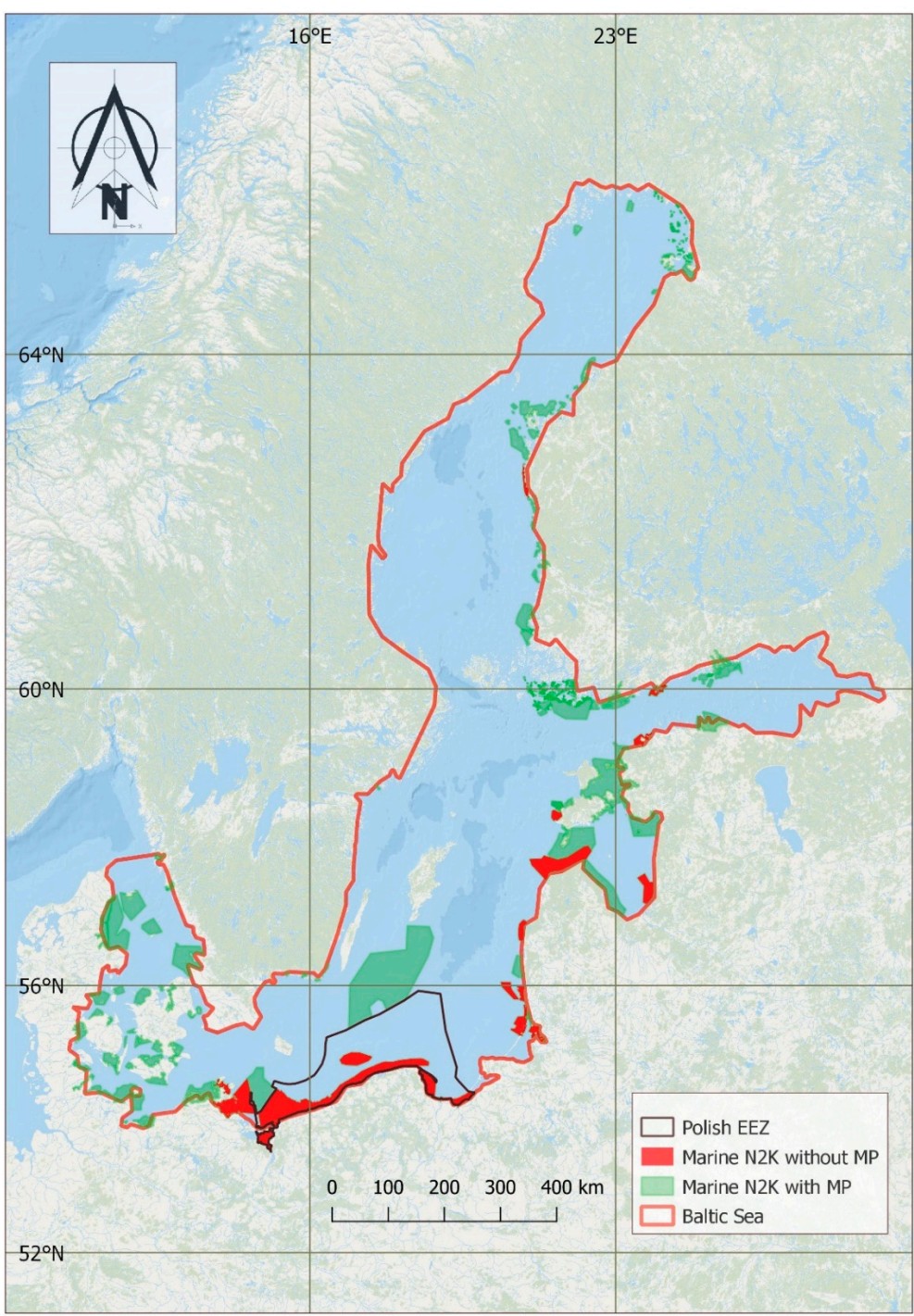

**Figure 1.** Study area—Baltic Sea, 117 Natura 2000 (N2K) areas established to protect waterbirds during their migration and wintering, with indication of which of them have a Management Plan (MP). Map created in QGIS ver. 3.4.8-Madeira (https://qgis.org/ accessed on 18 May 2019) under the GNU General Public License by Dominik Marchowski.

*2.4. The Exclusive Economic Zone with the Fewest Number of Management Plans Implemented*

More detailed analysis was carried out in the EEZ of the state with the smallest number of implemented marine SPA management plans. In this EEZ, we checked whether SPAs overlapped with Important Bird Areas of International Importance as designated

independently by BirdLife International [39] and whether MPs existed. In the absence of an MP for a given site, we investigated what was being done by state institutions for its implementation and how much money from public funds was being spent on it. Similar calculations, regarding population percentages and VFs for the entire Baltic, were carried out for the EEZ area with the weakest protection.

## 3. Results

### 3.1. Wintering Waterbirds on the Baltic Sea

Every year, 4,400,000 waterbirds spend the winter on the Baltic Sea: the most important group among them are seaducks (2,940,000—67%). Seventeen percent (17%) of the world's population of seaducks and thirty-five percent of Palaearctic seaducks winter in the Baltic Sea.

The Baltic Sea is crucial for the survival of the world's populations of long-tailed duck (LTD), velvet scoter (VS) and common scoter (CS), with up to 60%, 73% and 46% of these populations, respectively, wintering here. In the case of flyway populations, the importance of the Baltic Sea as a wintering ground is greater (the percentage of the flyway populations wintering in the Baltic Sea and the value factors are given in brackets): LTD (92.8%, $VF_{Baltic}$ = 204), VS (73.7%, $VF_{Baltic}$ = 153), greater scaup (GS; 74.1%, $VF_{Baltic}$ = 89) and CS (60.0%, $VF_{Baltic}$ = 55). The Baltic Sea is also important for the flyway populations of common and Steller's eiders (Figure 2).

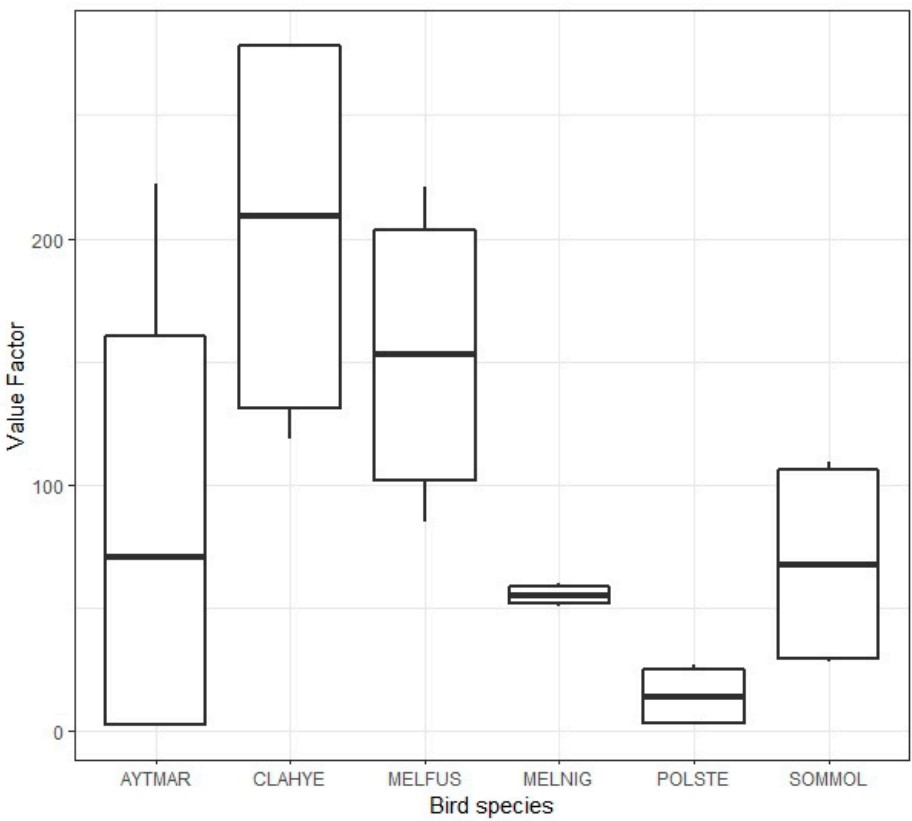

**Figure 2.** Value factors (VFs) for the most numerous waterbird (seaducks) species present during the non-breeding period in the Baltic Sea. Horizontal thick line—mean. Species abbreviations: AYTMAR—Greater Scaup, CLAHYE—Long-tailed Duck, MELFUS—Velvet Scoter, MELNIG—Common Scoter, POLSTE—Steller's Eider, SOMMOL—Common Eider.

The Special Protected Areas (SPAs) protecting VS and LTD within the EU are located almost exclusively in the Baltic Sea: 98% and 93%, respectively, of the EU population of these species winter there. There are MPs for 32 (65%) out of the 49 Baltic SPAs established

for the protection of VS, and for 15 (51%) out of the 29 SPAs designated for protection of LTD.

One hundred and seventeen (117) of the Baltic marine SPAs were classified as areas protecting waterbirds during the non-breeding period. These sites together cover an area of 54,177 km$^2$, which makes up 14% of the entire Baltic Sea. Ninety (90) of them (76%) have MPs. Deficiencies in MP implementation are not evenly distributed among the individual countries: Denmark—29 sites, 29 MPs implemented (100%); Sweden—12 sites, 12 MPs (100%); Finland—31 sites, 26 MPs (83%); Estonia—14 sites, 11 MPs (78%); Germany—12 sites, 9 MPs (75%); Latvia—5 sites, 2 MPs (40%); Lithuania—6 sites, 1 MP (17%); Poland—8 sites, no MPs (0%). The following analysis will refer to Poland, the only country where no MP has been implemented for any of its existing EEZ SPAs.

### 3.2. Wintering Waterbirds on the Polish Part of the Baltic Sea

A large part of the Baltic Sea is covered by the Polish EEZ (30,500 km$^2$) where, on average, 732,000 waterbirds (range 470,000–1,234,000) wintered from 2011 to 2018. Seaducks were the most numerous group of birds, making up on average 92.3% (mean from 2011–2018, range 85.8–94.0%) of all the waterbirds regularly present there. In 2017, 4.5% of the world's population of seaducks wintered in the Polish EEZ. This core group consists of four species: VS (up to 54% of the world's population, VU world), CS (up to 41% of the world's population, LC world), LTD (up to 17% of the world's population, VU world, and up to 33% of the flyway population, LC Europe) and GS (up to 2% of the world's population, LC world, and up to 63% of the flyway population, LC Europe) (Table 1).

**Table 1.** Numbers of waterbirds wintering in the Polish EEZ of the Baltic Sea compared to the numbers in the flyway and global populations. Values calculated based on the maximum numbers recorded in the Polish EEZ in the non-breeding period from 2011 to 2018.

| Species | Global Population | % in Polish EEZ | Flyway Population | % in Polish EEZ | IUCN Global threat Status | European Red List and Local Trend |
|---|---|---|---|---|---|---|
| Velvet Scoter | 451,500 [21] | 54 [32] | 450,000 [21] | 54 [32] | VU ↓ [38] | VU↓ [39] |
| Common Scoter | 687,000–815,000 [21] | 34–41 [33] | 687,000–815,000 [21] | 34–41 [33] | LC ? [38] | LC ? [39] |
| Long-tailed Duck | 3,200,000–3,750,000 [21] | 14–17 [32] | 1,600,000 [21] | 33 [32] | VU ↓ [38] | LC ↓ [39] |
| Greater Scaup | 4,760,000–5,095,000 [21] | 2 [40] | 150,000–275,000 [21] | 35–63 [40] | LC ↓ [38] | LC ↓ [39] |
| Red-breasted Merganser | 368,000–521,000 [21] | 1–2 [32] | 70,000–105,000 [21] | 6–10 [32] | LC → [38] | NT ↓ [39] |
| Goosander | 1,571,500–2,436,000 [21] | 2 [32] | 177,000–277,000 [21] | 13–21 [32] | LC ? [38] | LC → [39] |
| Smew | 99,000–123,000 [21] | 3–4 [32] | 24,000–38,000 [21] | 11–18 [32] | LC ↓ [38] | LC → [39] |
| Tufted Duck | 2,000,000–2,600,000 [21] | 2 [32] | 800,000–1,000,000 [21] | 5–6 [32] | LC → [38] | NT ↓ [39] |

The Polish part of the Baltic Sea is crucial for the survival of the global populations of VS (VF$_{Poland}$ = 113) and LTD (VF$_{Poland}$ = 49), and VF is also significant for the next six species (see Figure 3 and Table 1). Five SPAs were established within the Polish EEZ for the protection of VS and LTD (6500 km$^2$—21% of the EEZ). Up to 54% and 14% of the respective world populations of VS and LTD winter in this area. In some years, their flocks in Polish waters are extremely large, e.g., 73% of the EU population of VS in 2018 and 37% of the EU population of LTD in 2017.

Thirteen (13) marine Natura 2000 sites have been set up in the Polish EEZ, including eight SPAs and five SCIs. The area covered by the five SCI sites is 3600 km$^2$, i.e., 12% of the Polish seawater area, while that of the eight SPA sites is 7400 km$^2$, i.e., ca 24% of the Polish EEZ (Table 2). The SPA and SCI areas partially overlap. Ninety-four percent (94%) of the marine Important Bird and Biodiversity Areas in Poland have been designated as Natura 2000 SPA areas. Eight sites were designated in the same area as IBA, differing only slightly (see Table 3). The set of SPA sites lacks two IBA areas: East Border Waters (PLM4) and the Polish part of Southern Middle Bank—229 km$^2$ (SE067)—which lies mostly in Swedish waters (Table 3).

**Table 2.** List of Natura 2000 sites in Polish EEZ together with the area code and the legal basis on which the site was created (BD—Birds Directive, HD—Habitats Directive). MOG—Maritime Office in Gdynia, MOS—Maritime Office in Słupsk, MOSZ—Maritime Office in Szczecin. Numbers of implemented projects dedicated to the preparation of documentation for management plans of Marine Protected Areas, funds spent and the period of their spending.

| Site Code | Basis | Area km$^2$ | Date SPA/SCI Class | MP y/n | Managing Authority | No. of Project | Funds EURO | Duration |
|---|---|---|---|---|---|---|---|---|
| PLB220005 | BD | 625.2 | 2004 | no | MOG | POIS.05.03.00-00-281/10 | 700,889 | 2011–2014 |
| PLB280010 | BD | 322.7 | 2004 | no | MOG | POIS.05.03.00-00-281/10 | | |
| PLB220004 | BD | 17.5 | 2004 | no | MOG | POIS.05.03.00-00-281/10 | | |
| PLH220032 | HB | 266.0 | 2004 | no | MOG | POIS.05.03.00-00-281/10 | | |
| PLH220044 | HB | 8.8 | 2007 | no | MOG | POIS.05.03.00-00-281/10 | | |
| PLH280007 | HB | 409.2 | 2004 | no | MOG | POIS.05.03.00-00-281/10 | | |
| PLB990002 | BD | 1948.4 | 2004 | no | MOS | | 0 | |
| PLC990001 | BD/HB | 801.2 | 2004 | no | MOS | POIS.02.04.00-00-0027/17-00 | 891,691 | 2018–2020 |
| PLB320009 | BD | 471.6 | 2004 | no | MOSZ | POIS.05.03.00-00-280/10 | 422,626 | 2011–2014 |
| PLB320011 | BD | 125.0 | 2007 | no | MOSZ | POIS.05.03.00-00-280/10 | | |
| PLB990003 | BD | 3090.7 | 2004 | no | MOSZ | POIS.05.03.00-00-280/10 | | |
| PLH320018 | HB | 525.7 | 2006 | no | MOSZ | POIS.05.03.00-00-280/10 | | |
| PLH990002 | HB | 2429.5 | 2004 | no | MOSZ | POIS.05.03.00-00-280/10 | | |

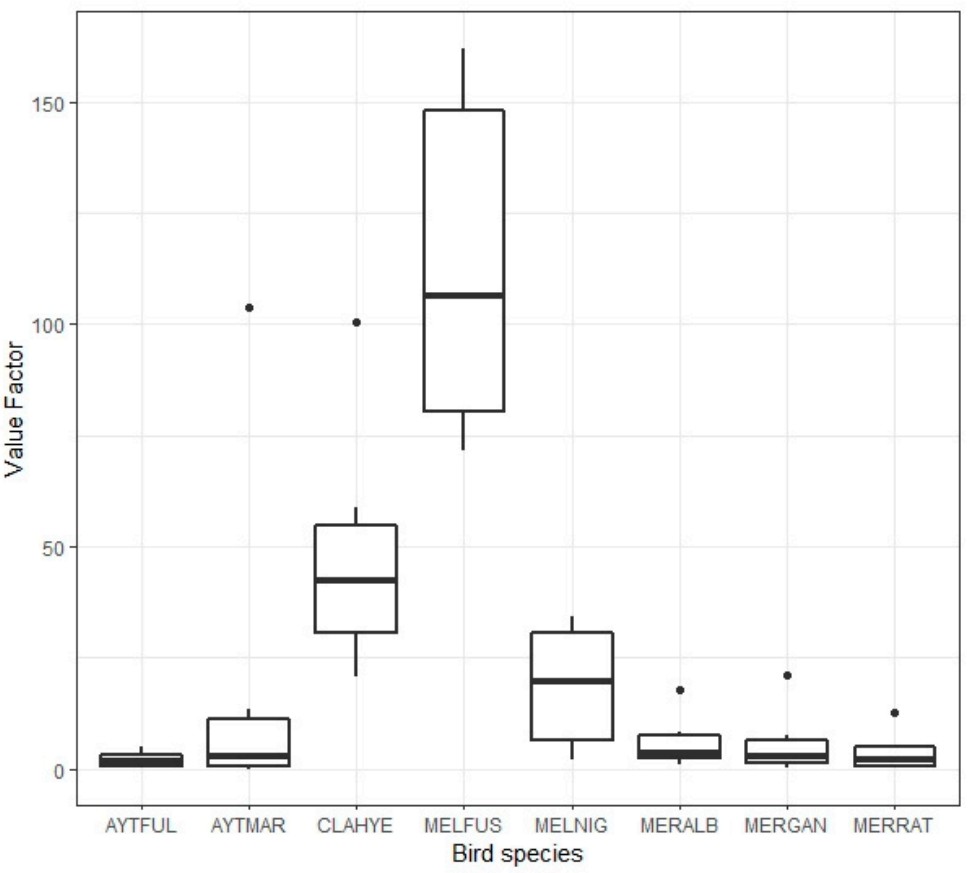

**Figure 3.** Value factors (VFs) for the most numerous waterbird species present during the nonbreeding period in Polish seawaters. Horizontal thick line—mean, dots—outliers. Species abbreviations: MELFUS—Velvet Scoter, CLAHYE—Long-tailed Duck, AYTMAR—Greater Scaup, MELNIG—Common Scoter, MERALB—Smew, MERGAN—Goosander, MERRAT—Red-breasted Merganser, AYTFUL—Tufted Duck.

**Table 3.** Comparison of the size of areas and their qualifying bird species in areas designated by BirdLife International (IBA) and by the Polish government (Natura 2000 SPA). Species abbreviations: VS—Velvet Scoter, LTD—Long-tailed Duck, GS—Greater Scaup, CS—Common Scoter, S—Smew, G—Goosander, RBM—Red-breasted Merganser, TD—Tufted Duck.

| IBA Code | SPA Code | SPA km$^2$ | IBA km$^2$ | SPA Qualif. sp. | IBA Qualif. sp. |
|---|---|---|---|---|---|
| PL002 | PLB320009 | 472 | 563 | TD, GS, S, G | TD, GS, S, G |
| PL011 | PLB320011 | 125 | 125 | S, G | S |
| PLM3 | PLB990003 | 3091 | 3119 | CS, VS, LTD, RBM | CS, VS, RBM, |
| PL024 | PLB220005 | 625 | 624 | TD, GS, VS, CS, S, G, RBM | TD, GS, S, G |
| PL029 | PLB280010 | 323 | 304 | TD, S | S |
| PLM2 | PLB990002 | 1948 | 1946 | VS, CS, LTD | LTD, VS |
| PLM1 | PLC990001 | 801 | 801 | LTD | LTD |
| PL027 | PLB220004 | 18 | 24 | TD, GS, LTD, S, G | G |
| PLM4 | - | 0 | 166 | - | VS |
| SE067 | - | 0 | 229 | - | LTD |

As of November 2022, no MPs have been implemented for any of the 13 Natura 2000 sites. Work on the preparation of MPs for 11 sites took place in 2011–2013 at a cost of EUR 1,123,000. Work on the MP project for the Słupsk Bank (PLC990001) area took place in 2018–2020; EUR 892,000 have been allocated for this purpose. No work has been completed for the preparation of an MP for the Coastal Baltic Sea Waters (PLB990002) (Table 2).

## 4. Discussion

The status of 43% of all seaduck species worldwide is threatened or near threatened [38]. A key area for the survival of this group of birds is the wintering grounds in the Baltic Sea, where 17% of the world's population of seaducks spend the winter. Eighty-five percent (85%) of EU seaduck species are threatened, near threatened, or declining. This demonstrates that, locally on the Baltic Sea, the conservation state of seaducks is poor. In the densely urbanised areas along the Southern and Western Baltic coasts, conflicts have arisen between the interests of sectors such as offshore wind farms, ship traffic or water sports and nature conservation objectives [23,41]. Although the bycatch threat is decreasing compared to previous decades because the fishing fleet is shrinking [26,42], it is still generating the highest mortality of diving waterbirds in the Baltic [43,44]. It is estimated that 76,000 waterbirds die in fishing nets every year on the Baltic Sea [25], which constitutes 19% of the total global bycatch in gillnets [44]. Such a high mortality rate combined with the large global populations of seaducks pose great challenges for countries with sea zones. Undoubtedly, a management plan is an effective means of conservation in these areas [29]. The implementation of MPs by EU Member States is an obligation enshrined in directives and transposed to the national laws of the MS (Council Directive 2009/147/EC).

In the 17 years since the Natura 2000 areas were established, not a single MP has been implemented in the Polish EEZ. In this respect, it is the only Baltic EU country that has no MP for marine Natura 2000 sites and is thus the weakest link in the system for protecting marine ecosystems in the Baltic Sea (excluding Russia, which is not a member of the EU). Poland has well-designated MPAs, which largely overlap with IBA sites (Table 3). Appropriate national laws were also introduced, which impose upon the managing authorities the obligation to implement MPs within six years of the site being established (Nature Protection Act, Journal of Laws 2018, item 1614). However, SPAs that are habitats of threatened species are in fact unprotected: without the legal regulation of conservation measures, they remain "Empty Shell Protected Areas". A country can list them in its statistics and can show what percentage of its surface area is protected, even though there is no real protection.

For several species, but especially VS and LTD, Polish seawaters are the most important wintering grounds in the world: neglecting to protect these areas will therefore affect their entire global population. One may speculate that the factor limiting the numbers of these

seaducks should be sought in these waters, where the greatest threat is posed by gillnet fishery bycatch [26,43,44].

The ban on fishing for Baltic cod (*Gadus morhua callarias*), introduced at the beginning of 2020 [45], will probably have a positive effect on reducing the bycatch of waterbirds. However, this is an additional benefit—an action aimed at protecting cod, so when the cod population has recovered, catches will resume, again threatening the birds. Gillnet fishing has not been banned in the lagoons, where species such as greater scaup *Aythya marila*, tufted duck *Aythya fuligula*, goosander *Mergus merganser* and smew *Mergellus albellus* continue to drown in the nets set for zander *Sander lucioperca* and bream *Abramis brama* [45].

In 2011–2013, the Maritime Office in Szczecin (managing the area on behalf of the Polish government) conducted an inventory of three marine SPAs: PLB320011 Kamień Lagoon and Dziwna, PLB320009 Szczecin Lagoon and PLB990003 Pomeranian Bay. At the same time, a similar project was conducted in the eastern part of the Polish marine area by the Maritime Office in Gdynia (Puck Bay PLB220005, Vistula River Mouth PLB280010 and Vistula Lagoon PLB220004). The projects were co-financed by the EU from the European Regional Development Fund (POIS.05.03.00-00-280/10, POIS.05.03.00-00-281/10). This work was the basis for drafting MPs. These documents were subjected to extensive public scrutiny [33–35]. Not only did they describe the importance of the area for birds, but they also identified threats and pressures to the designated conservation areas, made reference to existing spatial planning documents, specified measures for maintaining or restoring the appropriate conservation status of key species and indicated the most important conservation areas and recommendations for the birds occurring there [33–35]. Unfortunately, these MPs have not been implemented, even though 14 to 17 years have elapsed since those protected areas were established and despite the legal requirements in force in Poland to draw up MPs for them (Nature Protection Act, Journal of Laws 2018, item 1614). The EU encourages MSs to implement MPs by allocating funds for this purpose. Poland has used up some of these funds but has not implemented any MPs. Most of the projects aimed at establishing MPs finished a long time ago, but no analysis of the results has been carried out. *Value for money*, i.e., the scale of social benefits resulting from the money spent, has not been determined. However, this would have to be the subject of a separate analysis.

The failure to implement MPs results in the inability to effectively manage these conservation areas and the consequent deterioration of the state of protection of a number of them. Work is currently underway on Spatial Development Plans (SDP) for the Polish EEZ, which will reduce its functionality to six main activities: fishery, sport and recreation, transport, environmental protection, artificial islands, construction, and defence. In the proposals for these SDPs, less than 10% of the area has been designated for environmental protection, which is contrary to the EU Biodiversity Strategy for 2030 [11–13]. The earlier conservation proposals in the draft MPs have been ignored in SDP for the Polish EEZ (https://www.umgdy.gov.pl/ accessed on 9 September 2022; https://tinyurl.com/stqhgga accessed on 9 September 2022).

Recent scientific publications confirm that the most important threats identified in the draft MPs are still valid, e.g., bycatch, construction of marine wind farms or sporting and recreational activities [22,26,44]. The growing importance of the Southern Baltic for migrating and wintering birds [22,40,46] is associated with climate change and the shift in the range of the wintering areas of birds closer to their breeding grounds [47,48]. Local increases in bird numbers may then be perceived as a false improvement. In the complete absence of conservation measures and increasing threats, vulnerable species in these marine SPAs may be seriously endangered. The "ecological trap" phenomenon may be operating in these sites, and such "Empty Shell Protected Areas" may become low-quality "sink" habitats [49], thereby reducing the numbers of entire populations of waterbirds wintering in this part of Europe.

## 5. Conclusions

According to our analysis, the MPs for Polish MPAs have a scientific foundation, but they have not been implemented for political reasons. Unfortunately, EU regulations are not sufficiently stringent to convince Member States to implement them. We propose introducing a top-down EU regulation mechanism based on the assessment of priority species using VF. On this basis, it will be possible to identify gaps in the protection of the most threatened species and to implement conservation measures in the most appropriate sites. We believe that an effective way of getting Member States to implement MPs would be to make the receipt of EU funds dependent on the fulfilment of Natura 2000 obligations.

**Author Contributions:** D.M.: conceptualisation, investigation, methodology, formal analysis, visualisation, writing—original draft preparation; Ł.Ł.: conceptualisation, investigation, methodology, writing—reviewing and editing; J.K.: conceptualisation, investigation, writing—reviewing and editing. All authors have read and agreed to the published version of the manuscript.

**Funding:** This research received no external funding.

**Institutional Review Board Statement:** Not applicable.

**Data Availability Statement:** The study did not report any new data.

**Acknowledgments:** The authors thank the following people for sharing their photos used in the graphical abstract: Piotr Chara for the Long-tailed Duck and Miłosz Kowalewski for the Velvet Scoter.

**Conflicts of Interest:** The authors declare that they have no known competing financial interest or personal relationships that could have appeared to influence the work reported in this paper.

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
