# Peer review of "Management of Marine Natura 2000 Sites as Exemplified by Seabirds Wintering in the Baltic Sea: The Case of Poland"

_diversity, doi:10.3390/d14121081_

Round 1
Reviewer 1 Report
Review of the paper „Management of marine Natura 2000 sites as exemplified by seabirds wintering in the Baltic Sea: the case of Poland”
The article deals with establishing priority species of seabirds that can be used for monitoring and protection of all wintering seabirds in the Baltic Sea. I agree that Poland is an important country in this respect and that there is no official strategy for managing seabirds in the Baltic Sea. I suggest authors be more neutral in suggesting that someone is to blame for this situation, because it is too often commented in Europe that nature conservation in Poland is the subject of political party rivalry. Scientists must be neutral and cooperate with each party and only in terms of merit content. Besides, the manuscript is relatively well prepared, I only have a few minor comments.
I would like to draw the authors’ attention that their VF value factor is not innovative and I am curious if the authors are able to self-reflect on the assumption of VF. IUCN categories are based on the abundance trends and the population status of a species. I do not fully understand how the categories based on the number multiplied by the number of birds in a given location are supposed to help in the protection of species. The authors would have obtained a similar effect if they were working on the real abundances of species.
L13-14 – the wording in this sentence is not correct – I think it is not possible to decide which species is the most valuable from the protected species group, Please, rephrase – write, for example, that you select the species that is most useful as an indicator or define the priority species, as you wrote below.
L16 – please, explain VS and LTD – not all readers of the manuscript will be ornithologists
L20-22 – “Poland, a key country for the survival of these species, has not implemented a single MP despite the existence of documentation confirming their crucial importance for seaducks, and on the pressures occurring there.” – this is only the opinion of the authors, which the readers will not be able to verify. I suggest that you rephrase this sentence – it is usually a poor information system to blame in these cases. The authors also do not provide evidence that the documentation on the species' status is correct. This is not a sentence that should be the conclusion of an academic article. As a reviewer, I have not read this documentation and I cannot judge if the authors are right, while I have to evaluate the manuscript.
L36 – please, be objective and find discussions about how Natura 2000 works in Europe, e.g.
https://www.sciencedirect.com/science/article/pii/S0006320718312710
https://conbio.onlinelibrary.wiley.com/doi/pdf/10.1111/cobi.13434
https://www.sciencedirect.com/science/article/pii/S1470160X21011171
https://www.sciencedirect.com/science/article/pii/S1470160X21005744#s0075
L79-81 – please, rephrase. Birds gather on the sea like birds on the sea ;)
L108 – here is an explanation of an abbreviation that has been used earlier
L114-117 – please, rephrase. The space for specific hypotheses or scientific questions must not be a polite wish list of the authors.
L338-342 – this is not needed
L352-359 – please, rephrase; scientists should not be politicians
Author Response
Reviewer 1.
Review of the paper „Management of marine Natura 2000 sites as exemplified by seabirds wintering in the Baltic Sea: the case of Poland”
The article deals with establishing priority species of seabirds that can be used for monitoring and protection of all wintering seabirds in the Baltic Sea. I agree that Poland is an important country in this respect and that there is no official strategy for managing seabirds in the Baltic Sea. I suggest authors be more neutral in suggesting that someone is to blame for this situation, because it is too often commented in Europe that nature conservation in Poland is the subject of political party rivalry. Scientists must be neutral and cooperate with each party and only in terms of merit content. Besides, the manuscript is relatively well prepared, I only have a few minor comments.
[Autor response (hereafter AR): Thanks for your comments on our manuscript. Indeed, perhaps our text is a bit too biased, and we agree that it should be more neutral. We checked the entire manuscript in this respect and corrected it where necessary. Hopefully, our article is now more neutral.]
I would like to draw the authors’ attention that their VF value factor is not innovative, and I am curious if the authors are able to self-reflect on the assumption of VF. IUCN categories are based on the abundance trends and the population status of a species. I do not fully understand how the categories based on the number multiplied by the number of birds in a given location are supposed to help in the protection of species. The authors would have obtained a similar effect if they were working on the real abundances of species.
[AR: We realize that our VF proposal is not a great discovery but a simple multiplication of the IUCN threat status of a given species by the percentage of its abundance in a given area. This shows the importance of the site for a given species in a quantitative and not descriptive manner. It is a kind of message enhancement by multiplying the relative abundance with the threat status. Species with a high IUCN threat status and the highest relative abundance have the highest scores. The authors believe that such a method is simpler and perhaps more easily influencing the stakeholders and decision-makers. Hence, it may have an impact on decisions regarding the conservation of species and their habitats. Appropriate changes have been introduced to the text to describe our assumptions more clearly.]
L13-14 – the wording in this sentence is not correct – I think it is not possible to decide which species is the most valuable from the protected species group, Please, rephrase – write, for example, that you select the species that is most useful as an indicator or define the priority species, as you wrote below.
[AR: As suggested by the reviewer, we have rewritten this sentence.]
L16 – please, explain VS and LTD – not all readers of the manuscript will be ornithologists.
[AR: Yes, it is true, at the first mention in the text, for example, we wrote long-tailed Duck Clangula hyemalis - the full scientific and English name of the species and the abbreviation (LTD) in brackets, so that long species names are not repeated constantly in the text but use short abbreviations. To make it clearer, the word "hereafter" has been added to each of the first abbreviations given.
This also applies to other abbreviations such as Management Plan (hereafter MP).]
L20-22 – “Poland, a key country for the survival of these species, has not implemented a single MP despite the existence of documentation confirming their crucial importance for seaducks, and on the pressures occurring there.” – this is only the opinion of the authors, which the readers will not be able to verify. I suggest that you rephrase this sentence – it is usually a poor information system to blame in these cases. The authors also do not provide evidence that the documentation on the species' status is correct. This is not a sentence that should be the conclusion of an academic article. As a reviewer, I have not read this documentation and I cannot judge if the authors are right, while I have to evaluate the manuscript.
[AR: The authors do not fully agree with this comment. It is true that our text is not a typical scientific dissertation in which data is collected and then analysed and presented. Our text is somewhat of a review type, and we use sources that are to be verified. For example, the fact that Poland has not approved any Management Plan for marine Natura 2000 areas until the time of writing this article can be checked e.g. at: https://natura2000.eea.europa.eu/, there is a Standard Data Form (SDF) for each area, and in it, in item 6.2 there is a section on the Management Plan and this is where the information whether or not there is an MP. Therefore, it is not blaming someone but stating the fact - in Poland, there is not a single MP approved for marine Natura 2000 areas.
As for the confirmation and evidence that Poland is an important wintering area for the seabirds discussed in the article, they come from many sources. One of them is, for example, the already mentioned SDF, where the numbers of birds are given (although these data are often out of date, the authors used more recent, up-to-date data for the evaluation in this article). Other sources are publications cited in the article, Durinck et al. 1996, Skov et al. 2011, and Chodkiewicz et al. 2019. In the case of the last publication of Chodkiewicz et al. 2019, it presents the results of the state monitoring of wintering seabirds carried out annually since 2011, and one of the authors of this article is also the author of this publication, so he actively participated in the assessment of the number of seabirds in the Polish part of the Baltic Sea. The author is also a member of the HELCOM-OSPAR-ICES expert group - JWGBrird - which is an advisory body to the European Union on seabirds and activities under the Marine Strategy Framework Directive, so the authors are aware of all the latest information on abundance estimates at the biogeographic population level. Summing up, the data on the number of seabirds in the Polish part of the Baltic Sea comes from a reliable source, they were compared with the data of the entire population published by Wetland International at: https://wpe.wetlands.org/.]
L36 – please, be objective and find discussions about how Natura 2000 works in Europe, e.g.
https://www.sciencedirect.com/science/article/pii/S0006320718312710
https://conbio.onlinelibrary.wiley.com/doi/pdf/10.1111/cobi.13434
https://www.sciencedirect.com/science/article/pii/S1470160X21011171
https://www.sciencedirect.com/science/article/pii/S1470160X21005744#s0075
[AR: Thank you for your comments and interesting articles suggestions. As we mentioned earlier, we tried to rewrite the whole text to make it more objective.
We changed this sentence a bit to make it more objective and we cited one of the articles proposed by the reviewer.]
L79-81 – please, rephrase. Birds gather on the sea like birds on the sea ;)
[AR: Colloquial phrase "Like bees to a honeypot" has been removed.]
L108 – here is an explanation of an abbreviation that has been used earlier
[AR: Not entirely - it is more about the classification criteria for SPA areas and not about explaining the abbreviation. We tweaked the text a bit to make it clearer.]
L114-117 – please, rephrase. The space for specific hypotheses or scientific questions must not be a polite wish list of the authors.
[AR: Thank you for this comment, so we decided to delete the entire last sentence to be more objective.]
L338-342 – this is not needed
[AR: Thank you for this comment, so we decided to delete this paragraph as recommended by the reviewer.]
L352-359 – please, rephrase; scientists should not be politicians
[AR: Thank you for this comment, so we decided to delete the entire two last sentences to be more objective and less political.]
[To the above responses to the reviewer, I attach the manuscript in the change tracking mode with the introduced corrections.]

Reviewer 2 Report
The paper stresses on a simple definition or method (Value Factor) to highlight the importance of Nature 2000 areas in the EU; a definition which was not defined by the EU themselves. Focusing on the Baltic Sea area the authors show the importance of this area for many waterbirds and discusses their protection. Lacking any managements plans by the EU member state of Poland in these areas, the authors discuss the threads and pressures still occurring in this area. Hopefully this will result in the conservation of hundreds of thousands of especially Velvet Scoter, Common Scoter and Long-tailed Ducks for which Poland holds ~54% respectively ~41% and 17% of the world population!!
The paper is very well written and certainly needs attention to a broader audience.
A few minor details:
Ln 57-58: suggestion to rewrite “if this is not done, then in 2024 the EC will consider European enforcement legislation” by “otherwise the EC will consider European enforcement legislation by 2024”
Ln 79: remove “Like bees to a honeypot,”
Ln 111: suggest to rewrite “In our article” by “In this study”
Ln 220: replace seawaters with seas
Ln 237-238: do the authors mean “one” instead of “once”?
Ln 257: replace world’s by world
Ln 259: Wouldn’t compare the conservation status with the rest of the world. There are more areas not protected. Leave out “than overall in the world”
Ln 268: a bit of an awkward end of the sentence “zones in the sea”. Maybe “sea zones” is already a better option?
Ln 269: “Such areas” should be “these areas” or “EEZ”
Ln 291: replace "by the way" by “additional benefit”
Ln 352: replace “text” by “study”
Author Response
Rewiever 2
The paper stresses on a simple definition or method (Value Factor) to highlight the importance of Nature 2000 areas in the EU; a definition which was not defined by the EU themselves. Focusing on the Baltic Sea area the authors show the importance of this area for many waterbirds and discusses their protection. Lacking any managements plans by the EU member state of Poland in these areas, the authors discuss the threads and pressures still occurring in this area. Hopefully this will result in the conservation of hundreds of thousands of especially Velvet Scoter, Common Scoter and Long-tailed Ducks for which Poland holds ~54% respectively ~41% and 17% of the world population!!
The paper is very well written and certainly needs attention to a broader audience.
[Author response (hereafter AR): Thank you very much for the positive evaluation of our work. We improved our article according to the reviewer's suggestions below.]
A few minor details:
Ln 57-58: suggestion to rewrite “if this is not done, then in 2024 the EC will consider European enforcement legislation” by “otherwise the EC will consider European enforcement legislation by 2024”
[AR: Thank you for this comment, we have corrected this sentence as suggested by the reviewer.]
Ln 79: remove “Like bees to a honeypot,”
[AR: Removed as proposed by the reviewer.]
Ln 111: suggest to rewrite “In our article” by “In this study”
[AR: Thank you for this comment, we have corrected this sentence as suggested by the reviewer.]
Ln 220: replace seawaters with seas
[AR: We did not write "Polish seas" because it would suggest that there are at least a few Polish seas. To make it clearer, we wrote: "Polish part of Baltic Sea is crucial ..."]
Ln 237-238: do the authors mean “one” instead of “once”?
[AR: The point is that sometimes the IBA area is larger, and sometimes the SPA area is larger, the reader can see it in detail in Table 3.]
Ln 257: replace world’s by world
[AR: Corrected.]
Ln 259: Wouldn’t compare the conservation status with the rest of the world. There are more areas not protected. Leave out “than overall in the world”
[AR: Corrected.]
Ln 268: a bit of an awkward end of the sentence “zones in the sea”. Maybe “sea zones” is already a better option?
[AR: Of course, this is a better option – corrected.]
Ln 269: “Such areas” should be “these areas” or “EEZ”
[AR: Changed on “these areas”.]
Ln 291: replace "by the way" by “additional benefit”
[AR: Corrected.]
Ln 352: replace “text” by “study”
[AR: As suggested by the second reviewer, the entire last paragraph has been deleted.]
[To the above responses to the reviewer, I attach the manuscript in the change tracking mode with the introduced corrections.]
